# Associations of Age and Sex with the Efficacy of Inpatient Cancer Rehabilitation: Results from a Longitudinal Observational Study Using Electronic Patient-Reported Outcomes

**DOI:** 10.3390/cancers15061637

**Published:** 2023-03-07

**Authors:** Jens Lehmann, David Riedl, Alain Nickels, Gabriele Sanio, Marco Hassler, Gerhard Rumpold, Bernhard Holzner, Thomas Licht

**Affiliations:** 1Department of Psychiatry, Psychotherapy, Psychosomatics and Medical Psychology, University Hospital of Psychiatry II, Medical University of Innsbruck, 6020 Innsbruck, Austria; 2Ludwig-Boltzmann Institute for Rehabilitation Research, 1100 Vienna, Austria; 3Oncological Rehabilitation Center, 5621 St. Veit im Pongau, Austria; 4Paracelsus Medical University, 5020 Salzburg, Austria; 5Oncological Rehabilitation Center Sonnberghof, 7202 Bad Sauerbrunn, Austria; 6Evaluation Software Development (ESD), 6020 Innsbruck, Austria; 7Department of Psychiatry, Psychotherapy, Psychosomatics and Medical Psychology, University Hospital of Psychiatry I, Medical University of Innsbruck, 6020 Innsbruck, Austria

**Keywords:** health-related quality of life, cancer survivor, psycho-oncology, physical medicine, frailty, adolescents and young adults (AYA)

## Abstract

**Simple Summary:**

Cancer rehabilitation should restore patients’ quality of life (QOL), which is impaired by symptoms and treatment side effects. Here we investigate to which extent different age groups, frail patients, and men and women benefit from rehabilitation. To do this, reports given from patients themselves are used. We find that elderly patients suffer from a higher symptom burden and a lower QOL than younger individuals. Anxiety is more common among younger patients and women, while older patients tend to be more depressive. Regardless of age, sex and frailty, rehabilitation improves the QOL in these patient groups, and reduces distress and somatic symptoms.

**Abstract:**

Cancer rehabilitation is thought to increase the quality of life (QOL) and functioning of cancer survivors. It remains, however, uncertain whether subgroups benefit equally from rehabilitation. We wished to investigate the outcomes of multimodal rehabilitation according to age, sex and functioning. Patients of an Austrian rehabilitation center routinely completed the EORTC QLQ-C30 and the hospital anxiety and depression scale (HADS) questionnaires prior to (T1), and after rehabilitation (T2). To compare the outcomes between age groups (i.e., <40, 41–69, and ≥70 years), sex, and the Norton scale risk status, repeated measures of analyses of variance were calculated. A total of 5567 patients with an average age of 60.7 years were included, of which 62.7% were female. With T1 indicating the cancer survivors’ needs, older and high-risk patients reported lower functioning (all *p* < 0.001) and a higher symptom burden for most scales (all *p* < 0.05) before rehabilitation. Regardless of age, sex or risk status, the patients showed at a least small to medium improvement during rehabilitation for anxiety, depression, and most functioning and symptom scales. Some between-group differences were observed, none of which being of a relevant effect size as determined with the Cohen’s *d*. In conclusion, QOL is improved by rehabilitation in all patients groups, independently from age, sex, or the risk status.

## 1. Introduction

Advances in the management of cancer have resulted in improved 5-year survival rates for several cancer entities in Europe [1]. Not only cured patients, but also many individuals suffering from metastatic, incurable tumors can survive for extended periods due to progress in medical treatments [2]; thus, the number of cancer survivors is rising [3]. Many cancer survivors suffer, however, from adverse effects of surgery, chemotherapy or radiotherapy. Their QOL is impaired by pain, fatigue, nausea, cognitive impairments, vomiting, weight loss, loss of appetite, gastrointestinal symptoms, decreased muscular strength and endurance capacity, lymphedema or sleeping disorders [4,5,6,7]. Activities and participation can become permanently impaired, and some cancer survivors cannot resume their previous occupation [8].

The probability of developing a malignant tumor increases with age [2]. Older cancer patients have special requirements, and their needs may be different from those of younger patients [9]. They experience certain difficulties during cancer treatment, including a poorer tolerability for anticancer chemotherapy [10], a higher susceptibility for adverse events and complications, and a higher likelihood for a deterioration of their health status after treatment. Cancer itself, cancer treatment-related factors, or a combination of the two, can result in a functional decline, which appears to be accelerated in older patients with cancer [11]. Hence, the benefit of cancer treatments can differ, and patients with comorbidity or geriatric impairments are particularly at risk of poor health outcomes [12]. The population of older cancer patients is heterogeneous due to differences in functional capacity, and psychological and physical resources as well as comorbidity. In effect, a considerable number of elderly cancer survivors may eventually become frail or dependent on nursing care.

Frailty, i.e., a state of reduced physiologic reserve, is more prevalent in elderly than in younger persons. Frailty is considered as a phenotype, consisting of weakness, exhaustion, a slow walking speed, unintentional weight loss, and low physical activity [13]. This phenotype is associated with an increased likelihood of incident falls, worsening mobility or disability, hospitalization, and death [13]. Cancer treatment may increase the risk of becoming frail. Since frailty poses a risk for long-lasting immobility after injuries and falls, it can ultimately lead to pressure ulcers [14]. The risk for developing pressure ulcers can be estimated with the use of the Norton scale scoring system, which is widely used by nurses [15]. It includes the physical condition, mental state, activity, mobility and incontinence. The Norton scale has been identified as a predictor for QOL in elderly women [16]. Furthermore, the Norton scale scoring system has been used to assess the frailty of hospitalized patients and is a predictor of mortality in heart failure patients [17]. It is also predictive for complications during hospitalization other than pressure ulcers, and mortality in elderly patients admitted to an internal medicine department [18]. Low Norton scale scores are associated with falls in elderly patients with hip fractures, which may occur even long after a rehabilitation measure [19]. Thus, frail patients have special needs, which may be different from those of other patient groups.

Conversely, the needs of younger individuals may be quite different from those of middle-aged or elderly patients. The age group of young adults below 39 years represents a minority of cancer patients with a proportion of 3% [20,21]. The 5-year relative survival of adolescent and young adult (AYA) patients in both age groups, namely, 15–29 and 30–39 years, is higher or similar to that of older adults aged 40–49 [21]. The diagnosis and treatment of cancer, however, interrupts the emotional, social and physical development as well as the education and career perspectives of adolescent and young adult (AYA) patients. Every second patient of this group suffers from clinically increased anxiety, indicating a significant need for support [20]. The scores for general health, physical functioning, physical role limitations, and emotional role limitations are significantly worse for the majority of AYAs as compared with population norms [22]. AYA patients experience lower self-esteem, an altered sense of identity, fewer marital relationships, treatment-related sexual dysfunction, reproductive concerns and financial challenges [23]. For example, after hematopoietic stem cell transplantation, they reveal more requirements of psychological issues compared to non-AYA patients [24]. Furthermore, AYA and middle-aged adults have reported more financial distress than adults of 65 years and older [25]. Additionally, nutrition, body shape, sexuality and relaxation techniques are the most common psychological issues for AYAs [24]. AYAs often describe the need for more cancer-related medical and psychological information, and they perceive that cancer has had a negative impact on their control over their life, resulting in a lower health-related quality of life (HRQOL) [26]. It has, therefore, been highlighted that AYAs often face different HRQOL impairments than middle-aged or elderly patients [27,28]. This is reflected by the efforts to develop a core outcome set for the HRQOL of AYAs [29,30], which may include categories such as, the physical, cognitive, restricted activities, relationships with others, fertility, emotions, body image and spirituality/outlook on life [31]. In addition to their psychosocial distress, for example, young women under the age of 40 years who developed breast cancer experienced larger relative declines in HRQOL as compared with middle-aged and elderly women in physical roles, bodily pain, social functioning and mental health [32].

Moreover, sex differences have been reported with respect to unmet needs. Higher levels of psychological distress have been described in women as compared to men, and for some cancer types the prevalence may be two to three times higher than that seen for men [33]. In particular, female cancer patients suffer from a higher fear of progression than males [34,35]. Female survivors of colon cancer, for example, have reported greater problems completing daily activities due to physical problems and more pain than men [36]. In another study, however, men reported significantly more cancer-related impairments, more limitations in the activities of daily living, and poorer social resources than women [37]. This includes financial worries, which are higher in women than in men [25]. In a recent investigation, female cancer patients reported more symptoms and lower functioning scores in direct comparison to male patients [38]. In particular, they described more nausea, sleep disturbances, appetite loss, and the gastrointestinal symptoms of constipation and diarrhea than males in this investigation. In conclusion, although these reports were not fully consistent, sex differences in cancer survivors might possibly affect the degree to which rehabilitation can improve their differentially-impaired functions.

To mitigate the long-term effects of cancer and its treatment, cancer rehabilitation, also referred to as oncological rehabilitation, is deployed, whereby somatic, psychological and social consequences should be overcome in the best possible way [5,6]. Physical strength, endurance and impaired mobility should be enhanced, while pain control, a reduction in psychological distress, and an improvement in nutrition are equally important. Therapeutic procedures include physical treatments, emotional support, art and expression, psycho-educative lectures, lifestyle interventions, nutritional advice or smoking cessation [6,7]. Cancer rehabilitation is, therefore, conducted in a multidisciplinary way, which can be performed using an inpatient or an outpatient program.

There are different ways to assess whether or not cancer rehabilitation is effective in supporting patients in recovering physically and psychologically. Clinician-reported outcomes, observer-reported outcomes, performance outcomes, and patient-reported outcomes (PROs) can be used (as reviewed in Lehmann et al. [39]). PROs allow us to assess a patient’s own perception of their health status, independently from a third party’s interpretation. This is of particular interest because a concordance between a patient’s self-assessment and the clinical reports of many somatic symptoms is often inadequate [40,41,42,43]. Physicians tend to underestimate the psychological distress of cancer survivors, which can persist after the termination of the anticancer treatment [44,45,46]. It has been demonstrated that PROs capture the symptoms of cancer patients more accurately than assessments by physicians [47].

PROs have been utilized to analyze the efficacy of cancer rehabilitation. In an early study using self-assessment patient questionnaires, positive effects were reported in the somatic and psychosocial fields while the functional status remained unchanged [48]. German and Austrian trials on inpatient rehabilitation have demonstrated a reduction in anxiety and depression [49,50]. A three-month outpatient rehabilitation program in Belgium improved physical, emotional and role functions in comparison with a control group while social and cognitive functions remained the same [51]. Furthermore, a hospital-based rehabilitation program in North Carolina improved the HRQOL, fatigue, muscular endurance and flexibility [52].

We have implemented PROs into the clinical procedures of cancer rehabilitation, which help with understanding patients’ individual needs and organizing their therapeutic programs [53]. In view of the great differences among the various groups of cancer patients, we have questioned whether the therapeutic interventions currently in use are really appropriate for all or whether certain priorities must be set to overcome specific deficiencies.

We have previously investigated the needs of patient groups suffering from 21 different tumor entities [46]. We used baseline PROs before the rehabilitation to understand their still unmet specific needs, and a second evaluation with PROs at the conclusion of their rehabilitation to recognize the extent to which the rehabilitation met these needs. Thereby we have identified patients with cancers originating from lung, liver and esophagus as those with the lowest HRQOL. Furthermore, the physical function was particularly low in multiple myeloma, lung and liver cancer patients. A major finding of this study was the observation that the psychological distress of cancer patients was not necessarily associated with reduced physical functioning or a poor prognosis. Thyroid, lung and breast cancer were most burdened by anxiety while the highest levels of depression were noted in liver and brain cancer patients. Despite these differences we found that the oncological rehabilitation effectively improved the HRQOL, tumor-associated symptoms and psychological distress in all cancer entities [46].

These studies were performed without taking into account the age or the sex of the patients. In view of the particular needs of elderly individuals and of AYAs, we believed that it could not be assumed that cancer survivors of different age groups benefit from rehabilitative interventions in an identical way. Age is not a changeable factor and frailty is also difficult to influence, hence, these preconditions may be potential barriers to the efficacy of rehabilitation. The same is true for differences with respect to the needs of male and female patients. Thus, the implementation of rehabilitation must be adjusted towards adequately meeting these prerequisites.

The goal of the present investigation was, therefore, to identify the differences of HRQOL and functional health by age and sex, and to determine how rehabilitation might successfully restore impairments in these subgroups. Moreover, we wished to determine the extent to which frail patients with a risk of pressure ulcers would benefit from the treatment modalities we have used for rehabilitation. With the use of 15 scales of the EORTC QLQ-C30 instrument and the HADS, this investigation was aimed at identifying differences in the efficacy of the rehabilitative treatments in depth. To this end, we have further expanded the database we used in previous investigations [46], to allow for more comprehensive analyses.

## 2. Materials and Methods

### 2.1. Sample and Procedure

The data for this study are part of an ongoing clinical routine data collection at the Oncological Rehabilitation Center St. Veit im Pongau, Austria. The data collection and assessment of PROs have been described elsewhere in detail [54] and are summarized herein. The sample consisted of adult cancer survivors who had completed their active oncological treatment before referral to an inpatient rehabilitation measure. The rehabilitation program comprised 21 days of rehabilitation with 2–3 h of therapeutic units per working day. The rehabilitation could be extended for seven days in the case of a severe functional impairment of major psychological distress. The costs were covered by the Austrian pension funds, which require certain frequencies for the respective therapies as a basis for the treatment planning. According to the guidelines of the pension fund, patients had to be in a general condition that allowed them to actively participate in the therapeutic measures.

In this study we included all consenting patients undergoing rehabilitation measures between August 2014 and December 2019. In the case of repeated rehabilitation treatments, only data from the first rehabilitation stay were included in the study to avoid potential bias. Patients were excluded from the study if they (1) terminated the rehabilitation measure within the first three days; (2) had a prolonged interval (>56 days) between the completion of the T1 assessment and the start of the rehabilitation; or (3) had missing data for either the T1 or T2 assessment.

The first assessment of symptoms, functions and distress was conducted online via a patient portal prior to the initial admission (T1). The patients received login data to the portal, where they could complete the questionnaires. The computer-based health evaluation system (CHES) [55] was the basis for the patient portal and PRO assessments. At the T1, basic clinical data were assessed, including the cancer diagnosis; Karnofsky performance score (KPS); ECOG performance status scale; self-rated ability to work; and frailty (i.e., the risk of pressure ulcers) with the Norton score [15]. The HRQOL was assessed with the EORTC QLQ-C30 questionnaire, and psychological distress with the HADS. Results from the questionnaires were instrumental to plan the focus of therapies, and to help allocate resources ahead of the rehabilitation stay [53].

Once admitted to the rehabilitation program, the patients were asked to provide the written informed consent to participate in the observational study. If they agreed, they were included in the study for an evaluation of the treatment success. If they declined, their data were used merely for routine care and not included in the study. The second assessment (T2) was conducted at the end of rehabilitation with the same questionnaires as in the first assessment. The study had been reviewed by the Ethics Commission of the state of Salzburg (no. 415-EP/73/451-2014) and was conducted according to the principles of the Declaration of Helsinki.

### 2.2. Outcome Assessments

#### 2.2.1. EORTC QLQ-C30

The EORTC QLQ-C30 is a cancer-specific questionnaire, which comprises 30 items. It covers the domains of functioning (i.e., physical, social, role, emotional, and cognitive), the symptom burden (i.e., fatigue, nausea/vomiting, pain, dyspnea, sleep disturbances, appetite loss, constipation, diarrhea, and financial impact) as well as a global HRQOL scale. The scales were scored from 0 to 100, with 100 being the best score for the functioning scales, whereas 100 corresponds to the worst score for the symptom scales. Mean differences between the T1 and T2 were evaluated according to the evidence-based guidelines for interpreting change scores for the EORTC QLQ-C30 by Cocks et al. [56] and were classified as a trivial, small, and medium improvement following the classifications for each scale.

#### 2.2.2. Hospital Anxiety and Depression Scale (HADS)

The HADS is a 14 item questionnaire to assess psychological distress. It has two scales (i.e., anxiety, and depression), which are summed and range from 0 to 21. Typically, a result of 11 or greater for anxiety or depression is considered a clinically important result while scores between 7 and 10 are considered conspicuous [57]. The cut-off levels for relevant changes has previously been validated as 1.3 points for anxiety, and 1.4 points for depression [58]. We considered differences between the T1 and T2 as clinically meaningful if they were equal or greater than these differences.

#### 2.2.3. Norton Scale

The Norton scoring system has been used to evaluate frail patients’ risk of developing pressure ulcers. The score is an observer-based rating system based on five items on a 4-point scale to evaluate the patients physical condition, mental condition, activity, incontinence and mobility. A total score is calculated with higher scores indicating a higher risk for pressure ulcers in frail patients. A cut-off that has been defined to discriminate high- and low-risk patients has been introduced in previous research, with a total score < 15 points indicating a low frailty risk and a score ≥ 15 points indicating a high risk [15,59].

### 2.3. Statistical Analyses

Descriptive statistics of all included patients are shown. A Cronbach’s alpha for the HADS and the multi-item scales of the EORTC QLQ-C30 was calculated. Baseline differences (T1) between the groups (i.e., age group, sex, and frailty) were evaluated using a univariate ANOVA. To compare the patient outcomes based on age, sex and frailty, these variables were separately added as grouping variables in the ANOVA. The patients were assigned to three age groups: younger patients (<40 years); middle-aged patients (40–69 years); and elderly patients (≥70 years). The age up to which patients are considered AYA is not consistent in the literature, varying from 25 up to 39 years. While earlier articles have often used an upper limit of 25 years, the internationally and most broadly used definition more recently consists of an upper age limit of 39 years [24,26,29,38,60]. In accord with these studies, we have chosen to define young adults as those between the ages of 18 and 39.

Based on the above-mentioned cut-off of 15 points on the Norton score, patients were defined as low-risk or high-risk for frailty. For age, the Tukey HSD test [61] was used for pairwise comparisons of the <40 years and ≥70 years groups with reference to the 40–69 years group, as this group was considered the reference group from a clinical perspective, and differences between <40 years and ≥70 years were considered to be of a lower clinical relevance. Mean values for the HADS at T1 and T2 were compared to published reference values from the country’s general population [62].

Changes in the functional health (EORTC QLQ-C30 functioning scales), symptom burden (EORTC QLQ-C30 symptom scales), and anxiety and depression (HADS) between T0 and T1 were evaluated using repeated-measures analyses of variance (ANOVA). We included the factors time (T0 vs. T1) and group (e.g., age group; gender) and the interaction (time × group). To compare changes in the HRQOL during rehabilitation with regard to gender, we compared male and female patients using a repeated-measure ANOVA. To compare changes in the HRQOL during rehabilitation with regard to the Norton score risk assessment, it was necessary to include age as a covariate since the Norton score is associated with patient age. We ran repeated-measure analyses of covariance (ANCOVA) and separated the high and low risk patients using the cut-off < 15 points for the Norton score (Norton et al., 1962, Goldstone and Goldstone, 1988), while keeping age as a covariate (z-standardized). We included the factors time (T0 vs. T1) and group (Norton score high risk vs. low risk) and the interaction (time × group).

The *p*-values for all ANOVA and ANCOVA analyses were corrected to account for multiple testing using the Holm–Bonferroni sequential correction [63].

Changes in the HRQOL (EORTC QLQ-C30) and psychological distress (HADS) between the T1 and T2 were evaluated using repeated-measures analyses of variance (ANOVA). Since the Norton score is associated with patient age, we ran repeated-measure analyses of covariance (ANCOVA) for this factor, with the z-standardized age as a covariate. To determine the magnitude of the change during rehabilitation, we calculated the Cohen’s *d* for between-subject designs. The effect sizes were considered small for a *d* ≥ 0.2, medium for a *d* ≥ 0.5, and large for a *d* ≥ 0.8 [64]. Due to the large sample size of the study sample, there was an increased likelihood to find the statistically significant differences, which might not necessarily reflect the clinically-significant differences. We therefore defined a minimal important difference in the Cohen’s *d* according to the definition of a small effect ≥ 0.2 [64], to compare the effect sizes of improvement (within each group) across the different groups and, thus, to evaluate the clinical meaningfulness of found differences. The difference in the Cohen’s *d* is reported as *d^diff^*. Thus, only differences in the effect size between the groups (i.e., subtracting the effect size of group 1 from group 2) of the *d^diff^* > 0.2 points (20% of the pooled SD of both groups) were considered clinically relevant. SPSS, Version 26.0 (IBM Corp., Armonk, NY, USA) was used to conduct all analyses.

## 3. Results

### 3.1. Patient Characteristics

Data from a total of 7745 patients were initially available. Of those, n = 830 (10.7%) were excluded due to missing PRO data at T1 or T2, n = 938 patients (12.1%) were excluded due to repeated rehabilitation treatment, and another n = 410 patients (5.3%) were excluded since the interval between their T1 and the start of rehabilitation was longer than 56 days. The remaining n = 5567 patients were included in the analyses.

Most patients were middle-aged (70.5%), while about a quarter were elderly (25.0%) and the remaining 4.5% were younger patients. Of the younger patients (>40 years), 22 out of 248 were below 26 years of age. The majority of the sample was female (62.7%), which can be attributed to a high percentage of breast cancer patients (35.7%). Sex was most balanced in the ≥70 years group (45.0% male vs. 55.0% female). Almost 2/3 of the patients (62.5%) displayed a medium level of functioning, (i.e., 51–80% as rated by the KPS) and a grade 1 ECOG score (58.6%). The levels of functioning were lower in elderly patients, of whom only 26% were found with KPS levels > 80% as compared with 34.5% in the middle-aged group and 34.9% in the younger patients. Most elderly patients (72.2%) had KPS levels of >50% to 80%. Overall, 12.1% of the total sample were identified as high-risk frailty patients, with a substantially higher number among the elderly patients compared to the middle-aged or younger patients (29.4% vs. 6.8% vs. 1.9%, respectively). The most prevalent forms of cancer included breast, prostate, and colon malignancies. Prostate cancer was most frequent in the group of elderly men while no patient below 40 years suffered from this cancer. Conversely, testicular cancers were seen in younger and middle-aged, but not in older patients. Most cancer entities were noted among the patients of all age groups. The descriptive patient data are displayed in Table 1.

### 3.2. Treatment Modalities

Patients received multidisciplinary treatment in accordance with the guidelines of the Austrian pension fund, with at least 1800 therapeutic minutes within 21 days. An overview of the treatment modalities is displayed in Table 2. All the patients obtained treatment and guidance by physicians and nurses. Psycho-oncological counseling consisted of individual and groups sessions as well as relaxation exercises. According to the patients’ individual preferences, cognitive training, biofeedback, or sexual counseling could be included in the psychological counseling. The rehabilitative measures included individual and group physiotherapy, remedial massages, and medical training therapy consisting of aerobic and resistance training. In addition, the patients obtained social counseling, educational presentations including motivation to lifestyle modifications and nutritional advice. The majority of patients were also treated with occupational therapy, thermotherapy, hydrogymnastics, electrotherapy, or offered counseling for smoking cessation. Head and neck cancer patients obtained speech therapy and inhalation if deemed necessary. One quarter of patients were treated with manual lymph drainage for lymphedema. There was a tendency that the elderly patients were given more physiotherapy in groups while young patients participated more in resistance and aerobic training.

### 3.3. Internal Consistency

The Cronbach’s alpha was α = 0.82 and 0.83 for the HADS anxiety and depression scale, respectively. The Cronbach’s alpha for the multi-item scales of the EORTC QLQ-C30 ranged from α = 0.69 to 0.89 with lower alpha values for the scales with only two items.

### 3.4. Differences in Treatment Needs at Baseline Regarding Age, Sex and Frailty

At the baseline (T1), the whole sample consisting of rehabilitants of all age groups reported low global health/QOL with scores of 54.7 to 61.4 on a scale from 0 to 100. In contrast, an average of 75.65 has been described for the Austrian general population (Lehmann 2020). The elderly patients reported significantly worse physical functioning and global health/QOL than the middle-aged patients at baseline (T1) as well as higher levels on most symptom scales (*p* < 0.001); for details see Table 3. Additionally, patients with a high risk for frailty had significantly worse baseline scores (T1) for all functioning (all *p* < 0.001) and symptom scales (all *p* < 0.001). For details, see Table 4. As for sex-specific differences at the baseline, women reported significantly higher levels of social, emotional, and cognitive functioning and a better global health/QOL (all *p* < 0.05), while also reporting higher levels of pain, sleep disturbances, dyspnea, and constipation (all *p* < 0.05). Men on the other hand reported higher appetite loss and diarrhea (both *p* < 0.001). For details see Table 5.

### 3.5. Improvements in HRQOL and Psychological Distress

#### 3.5.1. Improvements by Age Group

By the end of the rehabilitation (T2), the global health/QOL was significantly improved to >74 points (younger and middle-aged patients), and to 70.7 points (elderly patients), respectively. Patients across all age groups reported statistically significant improvements during the rehabilitation for all EORTC QLQ-C30 functioning scales (all *p* < 0.001). The highest increases were observed for emotional functioning (improvement: 18.9–19.8 points), followed by social functioning (14.2–17.8 points) and role functioning (improvement: 12.8–15.1 points). The effect sizes for improvements in the functioning scales ranged from *d* = 0.17 (cognitive functioning) to the large effect sizes of *d* = 0.82 (emotional functioning) and *d* = 0.85 (global health/QOL). While there were significant time × group interactions for the scales of physical functioning and social functioning, all the between-group differences were *d^diff^* ≤ 0.15. The between-group differences were, therefore, considered of limited clinical relevance. Elderly patients started with significantly poorer physical functioning compared to the middle-aged group at T1, but the improvement tended to be even more pronounced among these patients (Table 3). There were no significant differences between the middle-aged and the younger patients with respect to the start of the rehabilitative measure and the improvement thereafter. As opposed to the other functions, the cognitive functioning was generally not very much impaired in all age groups; hence, the margin for improvement was lower.

As for the patients’ symptoms, a statistically significant reduction in all the assessed symptom scores was observed in the whole sample (all *p* < 0.001) and in each age group, with the largest improvements reported for fatigue (improvement: 15.5–16.9 points) and pain (improvement: 9.7–13.0 points). Patients reported improvements of at least small effect sizes (*d* > 0.2) in all the age groups for pain, appetite loss, constipation, and financial worries, and in the middle-aged and older patients for sleep disturbances and nausea. At the T1, the group of elderly patients reported the highest symptom burden, except for the scale for financial worries (Table 3). As compared with the middle-aged group, these differences were significant for fatigue, dyspnea, appetite loss, constipation and diarrhea; however, fatigue, nausea, sleep disturbances, loss of appetite, and diarrhea were particularly well reduced in these patients after the rehabilitation measure. The younger patients (<40 years) reported significantly more financial problems, which improved after the rehabilitation. We observed significant time × group interactions for the scales of pain, appetite loss, and sleep disturbances; however, since all the between-group differences were *d^diff^* ≤ 0.19, the between-group differences were considered to be of limited clinical relevance.

Figure 1 depicts the differences by age group in psychological distress as measured with the HADS at the T1 and T2. Our findings are shown in comparison to normative data of the general population (Hinz and Brähler, 2011). The age groups significantly differed for both anxiety and depression at T1 (Table 3). Both anxiety and depression significantly improved over time (i.e., the mean effects of time were both *p* < 0.001). Across all the age groups, the patients reported improvements well above the cut-off levels for clinical significance as defined by Puhan (2008) [58]. There was a significant interaction (time × group) for anxiety, indicating that groups improved statistically-significantly different from each other; however, the between-group difference in *d* was <0.2 (*d^diff^* = 0.10) and was, therefore, considered to be of a lower clinical relevance. Interestingly, at T2, the levels of depression were reduced in all the age groups even below those found in the general population (Figure 1). The effect sizes for improvement were of a medium size and ranged from *d* = 0.44 (anxiety) to *d* = 0.60 (depression).

#### 3.5.2. Improvement by Norton Scale Risk Score

The effect sizes for an improvement in the functioning scales ranged from *d* = 0.16 (cognitive functioning) to the large effect of *d* = 0.87 (global health/QOL), and in the symptom burden from *d* = 0.07 (dyspnea) to *d* = 0.69 (fatigue). In the functioning scales, we observed statistically significant time × group interactions for the scales of physical functioning and social functioning with high-risk patients showing larger improvements; however, all the between-group differences were *d^diff^* ≤ 0.11 and were thus considered to be of limited clinical relevance. In terms of the symptom scores, the high-risk patients showed significantly larger improvements for nausea/vomiting and dyspnea; however, since none of the mean difference of effect sizes exceeded 0.2 (all *d^diff^* ≤ 0.11), they were considered to be of limited clinical relevance. For details see Table 4.

Table 4 also shows the improvements in psychological distress at T1 and T2. Both the high- and low-risk patients significantly differed for both anxiety and depression at T1, with the high-risk patients showing higher levels of psychological distress. Both anxiety and depression were improved over the above-mentioned cut-off levels for clinical significance (Puhan, 2008) [58]. The effect sizes for improvement ranged from *d* = 0.44 (anxiety) to *d* = 0.60 (depression). The interaction of the time × group was not significant for depression nor for anxiety, indicating that the benefit for both groups was statistically similar.

#### 3.5.3. Improvements by Sex

Finally, we investigated the sex-specific effects of cancer rehabilitation in male and female patients. Both males and females reported significant improvements across all the functioning and symptom scales, with the most pronounced effects for emotional functioning (improvement: 17.3–20.5 points) and social functioning (improvement: 15.4–15.7 points), as well as fatigue (improvement: 14.9–16.0) and pain (improvement: 10.5–10.6 points). We observed significant time × group interactions for emotional functioning, indicating that the female patients improved more in emotional functioning; however, the between-group difference in *d* was <0.2 (*d^diff^* 0.12) and was, therefore, considered to be of a lower clinical relevance. There were no significant interactions (time × group) for any of the EORTC QLQ-C30 symptom scales, indicating that the improvement in the symptom burden was similar for both men and women. For details, see Table 5.

We observed differences with regard to psychological distress (Table 5). At T1, the female patients reported significantly more anxiety compared to the male patients before starting the rehabilitation. From T1 to T2, anxiety and depression were both markedly decreased in men and in women to a clinically meaningful extent (Puhan 2008) [58]. The improvement in anxiety was particularly noticeable in the female patients while depression was reduced in a similar fashion in both sexes. There was a significant interaction (time × group) for anxiety, indicating that men and women improved significantly different from each other; however, the between-group difference in *d* was <0.2 (*d^diff^* = 0.1) and was, therefore, considered to be of a lower clinical relevance.

## 4. Discussion

The aim of our study was to determine differences in the HRQOL and functional health between cancer survivors of different ages and sexes, and to identify to which extent impairments could be restored by rehabilitation. To this end, we used PROs collected in the routine care of patients in cancer rehabilitation, which can be considered the gold-standard when assessing patients’ functional health. To the best of our knowledge and as of 2023, this is the largest study that has analyzed and compared the efficacy of rehabilitation in different subgroups of cancer survivors.

### 4.1. Patient Group Differences in HROQL at the Start of Rehabilitation

We found that in each age group studied, regardless of the risk of pressure ulcers in frail cancer survivors, and in men and women, the HRQOL in patients before the start of rehabilitation was lower compared to the sex and age-matched data from the general population in Austria [65].

There were also notable significant differences between the patient groups at the start of rehabilitation. Regarding sex, there were several significant differences in the HRQOL and psychological distress at the T1. Compared to men, women reported significantly lower emotional functioning. On the symptom scales, women reported a significantly higher symptom burden for most scales (such as fatigue or sleep disturbances), but also a lower symptom burden on two scales (i.e., appetite loss and diarrhea). These differences are in line with differences also found in the Austrian general population (e.g., emotional functioning, fatigue, and diarrhea) [65]. Women also reported significantly higher anxiety scores compared to men, but there were no differences with regard to depression. This finding is different from reports that age, gender, marital status, social class and education are not associations consistently seen with anxiety in cancer patient populations [66]. A direct comparison had indicated that anxiety, depression, fatigue and dyspnea were more frequent in male cancer survivors than in females, whereas female patients suffered more from nausea and vomiting, insomnia, appetite loss, constipation and diarrhea [67]. In particular, they described more nausea, sleep disturbances, appetite loss, and the gastrointestinal symptoms of constipation and diarrhea than males in this investigation; however, fatigue and dyspnea as well as anxiety and depression, were reported more frequently in the male patients than in the female ones when adjusted to an age-matched reference population. In this study, males also displayed a more significant net loss in the role and social functioning than females when compared to a reference population.

On the contrary, Hinz et al. [68] have described higher levels of a fear of progression in female patients than in males. Our observation of higher levels of anxiety in female patients is presumably related to the high level of anxiety in female breast cancer patients, which we have reported before [54]. Breast cancer patients accounted for 35% of patients in our current study. Similar to reports from colon cancer patients [36], the women in our sample reported significantly higher levels of pain compared to men, but not poorer physical functioning.

Regarding age, we found several significant differences between the groups. Contrary to what one might expect, most self-reported impairments of AYAs (the youngest group of patients < 40 years of age) were quite similar to those of middle-aged patients. In particular, the anxiety and depression of AYAs were not significantly different from the middle-aged group. Conversely, there were no group differences in EORTC QLQ-C30 emotional functioning. In a Canadian study on AYA cancer survivors [69], AYAs experienced a significantly higher risk of psychosocial distress than older adult survivors. In our study, the most notable difference between the AYA and older patients concerned the financial impact of the disease. This may be attributed to work and employment interruptions, hindered professional achievement or maintained financial dependence from parents, and barriers to adequate insurance coverage [70]. It should be noted that a minority of the group of younger patients was below the age of 26 years in our study (22 of 248 patients); thus, the needs of the youngest group of adult cancer patients may be represented to lesser degree. We have recently reported on the needs and outcomes in the cancer rehabilitation of children and juvenile AYAs below the age of 18 [71]. Pediatric rehabilitation is being performed in a separate department of our rehabilitation center under different guidelines from the Austrian health insurance, thus, the two groups of AYAs below and above the age of 18 could not be combined in this investigation.

Older adults described significantly fewer financial concerns, probably due to broad coverage by the Austrian pension fund. The financial hardship of older cancer survivors is presumably more pronounced in countries with a different social system [72], while Austria’s residents report the lowest levels of unmet needs for medical care across the European Union [73]. Elderly cancer survivors (≥70 years) described a significantly lower quality of life than middle-aged individuals (40–69 years) on many scales. For example, they reported lower physical functioning, more fatigue, appetite loss, constipation, diarrhea, and dyspnea, but less anxiety and depression than middle-aged patients. By identifying the age-dependent needs of cancer survivors, our study may help design more specific treatment programs for them.

We also found that patients who were at a high risk of pressure ulcers, compared to patients at a lower risk, reported a significantly worse HRQOL on all scales of the EORTC QLQ-C30, with the highest point differences in the functional health and symptom burden for the physical functioning (23.2 point difference) and fatigue scale (15.8 point difference), respectively. This indicates a generally poor clinical condition of this patient group. Moreover, the high-risk patients also reported significantly higher psychological distress at T1 compared to the low-risk patients. Importantly, as age was a covariate in the analyses, these differences indicate that the risk of pressure ulcers is, independently of age, associated with a lower HRQOL and a higher psychological distress. The Norton score may be considered a surrogate measure for aspects of frailty [74] or even mortality [75]. Therefore, our findings do not come as a surprise, as patients who are considered frail often exhibit a worse HRQOL compared to non-frail patients [76,77,78].

### 4.2. Improvements in HRQOL during Rehabilitation in All Groups and Similar Improvement in All Groups

In our previous study, we found that all patients’ HRQOL and psychological distress generally improved during cancer rehabilitation [54]. Following this finding and considering the differences found at the start of rehabilitation, we investigated whether there were relevant group-level differences in *how much* and *which* groups improved during the rehabilitation measure.

The most striking improvements during rehabilitation were observed for emotional functioning, with large effect sizes in all age groups, in women, and in patients both at a high as well as a low risk for pressure ulcers. In addition, depression, fatigue, and, with one exception, social functioning were improved with at least medium-sized effects in all the patient groups. Our finding is in accord with an individual patient data meta-analysis, which confirms that psychosocial interventions improve the QOL, emotional and social function in patients with cancer [79]. The data presented here suggest that psychosocial counseling and support are effective in all age groups and for both sexes. The treatment of cancer-related fatigue is commonly performed with psychological interventions and with exercise, which have both revealed an effectiveness to overcome this condition in meta-analyses [80,81]. Thus, cancer rehabilitation is generally performed by a multimodal treatment consisting of supervised exercise and medical training in combination with psychological counseling units [5,6]. Exercise exerts additional positive effects, consequently alleviating depressive symptoms and anxiety [82,83].

An important finding is that frail patients with a high risk of pressure ulcers benefitted from rehabilitation. They reported significant improvements in their global health/QOL and social, role and emotional functions. The improvement of the physical function of elderly and frail patients are mainly attributed to physical and occupational therapies [84] including resistance and endurance training [85]. In fact, we had doubted that, in the absence a comprehensive geriatric assessment (CGA), frail patients would benefit sufficiently from the rehabilitation, and would need specific geriatric rehabilitation measures. Our routine assessment covered several components of CGA. In a meta-analysis of studies on the CGA used in studies on elderly cancer patients [86], all trials used the ECOG performance score, and identified comorbidities and the activities of daily living (ADL) index. Furthermore, clinical tools for assessing depression, mobility, nutritional status, and cognitive dysfunction were used in the majority of studies [86]. Despite the observed improvement of frail patients, the incorporation of CGA into the rehabilitative program of elderly and frail patients is strongly recommended.

With regard to the question of whether groups show a significantly different change over time in the HRQOL during rehabilitation, we found several statistically significant interactions of the time × group for improvements in functional health, symptom burden, and psychological distress. For example, the improvements in physical and social functioning, and on the symptom scales for nausea/vomiting and dyspnea were significantly higher in the high-risk group compared to the low-risk group; however, all the between-group differences in effect sizes were smaller than the threshold for a small effect size as specified by Cohen [64]. This indicates that, while some groups tended to profit statistically significantly more than others did, the differences in improvement between the different groups (for age, sex, or Norton score risk assessment) were overall of a lower clinical relevance. Thus, our findings rather indicate that the improvement was remarkably similar regardless of the patients’ age group, sex, or their risk for pressure ulcers, and that all patients profited from the rehabilitation measure. With only a single exception, for all the groups and for all scales, at least moderate improvements according to the thresholds defined by Cocks et al. [56] for the EORTC QLQ-C30 and by Puhan [58] for the HADS were found. The results of our study thus indicate that patients from all subgroups substantially profit from inpatient rehabilitation; however, given the baseline differences especially across the age groups in our sample, our results also highlight the need for subgroup-specific treatment focuses. Elderly patients, for example, show higher rates of physical impairment and might, therefore, profit from a stronger focus on physiotherapy, while younger patients seem to struggle more with financial difficulties and might profit from a stronger support by a social worker. This is a good practical example of how the results of our study can directly influence the quality of care in rehabilitation settings, for at the moment, all patients receive the same amount of counseling by social workers during rehabilitation. Based on the results of our study, we plan to increase the amount of counseling by social workers for younger patients, to improve their rehabilitation outcomes. Furthermore, elderly patients suffer more than younger ones from a loss of appetite, constipation or diarrhea; thus, they would benefit from more advice by dietologists. In addition, while all the function and symptom scores were worse in frail patients with a high risk of pressure ulcers, special attention should be paid to the level of depression, which was higher than observed in any other subgroup; thus, the amount of psychological support should be increased for these patients. We intend to take those findings into account when allocating treatment resources. Future studies should focus on specific treatment arms for patients in different age groups.

Combined, these findings indicate that all groups, regardless of age, sex or risk score profit from cancer rehabilitation similarly and show at least a small and non-trivial improvement. For the functional domains, which are at the core of rehabilitation (i.e., restoring patients functioning), high improvements were found which were often considerably above the threshold for medium improvement as specified by Cocks et al. [56]. Unfortunately, the guidance by Cocks et al. provides “at least medium“ as a maximum improvement; therefore, following this definition, we cannot specify if the improvements were possibly larger than medium size.

### 4.3. Strenghts and Limitations

Our study has several strengths and limitations. One strength is the use of data from clinical routine procedures over a period of more than four years, which provided us with a very large database and enabled us to perform various subgroup analyses. Secondly, included this study were unselected adult cancer patients suffering from any cancer entity. We believe that this study represents the real-life data of cancer survivors’ needs as analyzed before the start of the rehabilitation. Thirdly, a good compliance of patients with data acquisition was achieved with a high percentage of completed questionnaires. The return rate for both the T1 and T2 time-points was 84.0%, and even older patients had high participation rates.

One major limitation is that it was a single-center study. A generalization of the results should be applied with caution. Nevertheless, our results on the whole patient cohort are well comparable to those reported by Klocker et al. [49], in which a lasting improvement after the end of the rehabilitation measure was documented. Secondly, we used only instruments for general QOL, functions and symptoms. The use of instruments for specific tumor entities was not feasible because the initial pilot studies had revealed that many patients were less cooperative if too many questions had to be answered. Furthermore, the study was limited by its observational nature and the fact that no control group was available. Patients are referred to the rehabilitation center by the Austrian pension fund, thus, a randomization could not be performed. Thirdly, we did not compare the subgroups by cancer type as this would further split up the sample (e.g., the number of patients with brain cancers below 40 years was only 13); however, our previous publication [54] gives some indication of how the HRQOL differs between diagnoses. Finally, we note that there are no agreed standards on how to best compare differences in within-group change in HROQL at the between-group level. Due to our large sample size, we had a higher probability of finding statistically significant, but not necessarily meaningful differences. We therefore relied on a minimal important difference derived from Cohen [64]; in our analysis, we argue that any difference in the Cohen’s *d* (*d^diff^*) smaller than a small effect size of *d* = 0.2 does not indicate a meaningful difference between the groups. This approach allowed us to identify meaningful differences, not merely statistical significance. There is, however, currently no generally-accepted and validated methodology for this purpose.

## 5. Conclusions

We identified marked differences between cancer survivors of different age or sex with respect to anxiety, depression and symptom burden. Nonetheless, cancer rehabilitation increased the QOL in all the investigated groups in a similar fashion, due to an improvement in functioning and a reduction in somatic symptoms and psychological distress. Our study may help design rehabilitative programs according to the specific needs of different groups of cancer survivors.

## Figures and Tables

**Figure 1 cancers-15-01637-f001:**
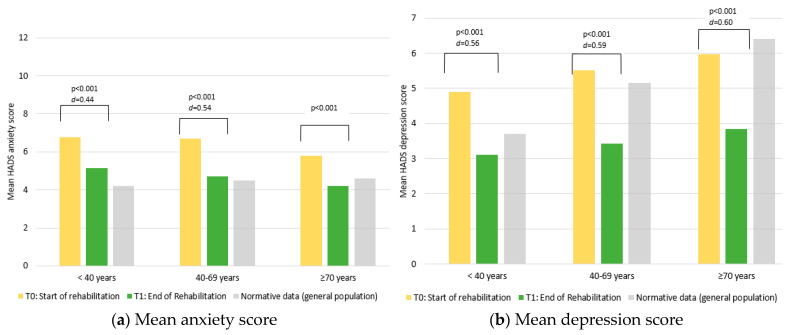
Mean HADS anxiety and depression scores (**a**,**b**), stratified by age groups, compared to data from the general population [62].

**Table 1 cancers-15-01637-t001:** Sociodemographic and clinical characteristics.

	Total Sample	>40 Years	40–69 Years	≥70 Years
	N (%)	N (%)	N (%)	N (%)
Sample size (row %)	5567	248 (4.5)	3927 (70.5)	1392 (25.0)
Sex				
Male	2077 (37.3)	87 (35.1)	1363 (34.7)	627 (45.0)
Female	3490 (62.7)	161 (64.9)	2564 (65.3)	765 (55.0)
Age				
Mean (SD)	60.7 (12.0)	33.1 (4.8)	56.9 (7.0)	76.2 (4.6)
Body mass index (BMI)				
Mean kg/m^2^ (SD)	25.8 (5.2)	24.4 (5.9)	25.8 (5.3)	26.0 (4.7)
Smoker	936 (16.8)	60 (24.2)	760 (19.3)	116 (8.3)
KPS				
High level of functioning (81–100%)	1805 (32.7)	109 (43.9)	1337 (34.5)	359 (26.0)
Medium level of functioning (51–80%)	3675 (65.2)	136 (54.8)	2532 (65.3)	1007 (72.9)
Low level of functioning (0–50%)	111 (2.6)	3 (1.2)	12 (0.3)	14 (1.0)
Missing information *	58	0	46	12
ECOG score				
Grade 0	417 (7.6)	31 (12.6)	324 (8.4)	62 (4.5)
Grade 1	3219 (58.6)	167 (67.6)	2400 (62.1)	652 (47.3)
Grade 2	1787 (32.5)	45 (18.2)	1115 (28.8)	627 (45.5)
Grade 3	70 (1.3)	4 (1.6)	28 (0.7)	38 (2.8)
Missing information *	74	1	60	13
Norton scale (n = 4791, 84.8%)				
Mean	16.7 (1.9)	18.2 (1.3)	17.1 (1.6)	15.4 (2.0)
<15 points (high risk)	569 (12.1)	4 (1.9)	228 (6.8)	337 (29.4)
≥15 points (low risk)	4150 (87.9)	212 (98.1)	3127 (93.2)	811 (70.6)
Norton scale not assessed *	848	32	572	244
Cancer entities				
Head and neck cancers (C00–14; C30–C32)	285 (5.1)	3 (1.2)	229 (5.8)	53 (3.8)
Esophageal cancer (C15)	75 (1.3)	2 (0.8)	50 (1.3)	23 (1.7)
Gastric cancer (C16)	149 (2.7)	1 (0.4)	89 (2.3)	59 (4.2)
Colon cancer (C18–19)	327 (5.9)	3 (1.2)	218 (5.6)	106 (7.6)
Rectal cancer (C20–21)	184 (3.3)	1 (0.4)	127 (3.2)	56 (4.0)
Liver cancer(C22)	38 (0.7)	0 (0)	16 (0.4)	22 (1.6)
Pancreatic cancer (C25)	118 (2.1)	1 (0.4)	67 (1.7)	50 (3.6)
Lung cancer (C33–C34)	273 (4.9)	3 (1.2)	164 (4.2)	106 (7.6)
Skin cancer (C43–44)	50 (0.9)	2 (0.8)	31 (0.8)	17 (1.2)
Breast cancer (C50)	1965 (35.3)	66 (26.6)	1565 (39.9)	334 (24.0)
Uterine cancers (C51–55)	182 (3.3)	11 (4.4)	134 (3.4)	37 (2.7)
Ovarian cancer(C56)	200 (3.6)	7 (2.8)	145 (3.7)	48 (3.4)
Prostate cancer (C61)	441 (7.9)	0 (0)	274 (7.0)	167 (12.0)
Testicular cancer (C62)	58 (1.0)	23 (9.3)	35 (0.9)	0 (0)
Renal cancer (C64)	115 (2.1)	1 (0.4)	80 (2.0)	34 (2.4)
Bladder cancer (C67)	122 (2.2)	2 (0.8)	67 (1.7)	53 (3.8)
Brain cancers (C70–72)	77 (1.4)	13 (5.2)	54 (1.4)	10 (0.7)
Thyroid cancer (C73)	52 (0.9)	14 (5.6)	35 (0.9)	3 (0.2)
Malignant lymphomas (C81–C86; C88)	369 (6.6)	65 (26.2)	234 (6.0)	70 (5.0)
Multiple myeloma (C90)	88 (1.6)	0 (0)	60 (1.5)	28 (2.0)
Leukemias (C91–C95)	108 (1.9)	17 (6.9)	64 (1.6)	27 (1.9)
Other cancer types	291 (5.2)	13 (5.2)	189 (4.8)	89 (6.4)

KPS: Karnofsky performance score; ECOG: Eastern Cooperative Oncology Group. * Missing data were not included in the calculation of percentages.

**Table 2 cancers-15-01637-t002:** Overview of treatment modalities across age groups per patient during rehabilitation.

	Total Sample				>40 Years				40–69 Years				≥70 Years			
	N	%	mdn	IQR	N	%	mdn	IQR	N	%	mdn	IQR	N	%	mdn	IQR
Guidance and treatment by physician	5567	100.0	6	5–7	248	100.0	6	5–7	3927	100.0	6	5–7	1392	100.0	6	5–7
Psycho-oncological counseling (group)	5567	100.0	1	1–1	248	100.0	1	1–1	3927	100.0	1	1–1	1392	100.0	1	1–1
Relaxation therapies	5566	99.9	3	3–5	248	100.0	4	3–5	3927	100.0	3	3–5	1391	99.9	4	3–5
Physiotherapy (individual treatment)	5565	99.9	6	5–8	248	100.0	6	5–8	3926	100.0	6	5–8	1391	99.9	6	5–8
Nutritional advice	5563	99.9	4	3–4	248	100.0	3	3–4	3925	99.9	4	3–4	1390	99.9	4	3–4
Psycho-educative lectures	5558	99.8	3	2–3	247	99.6	3	2–4	3921	99.8	3	2–3	1390	99.9	3	2–3
Nursing procedures	5555	99.8	2	2–2	248	100.0	2	2–2	3917	99.7	2	2–2	1390	99.9	2	2–2
Physiotherapy (groups)	5534	99.4	7	5–10	244	98.4	6	5–8	3910	99.6	7	5–9	1380	99.1	8	6–11
Medical training therapy—aerobic training	5506	98.9	7	5–9	246	99.2	7	6–9	3894	99.2	7	5–9	1366	98.1	6	5–9
Psycho-oncology (individual counseling) including biofeedback	5361	96.3	5	4–7	239	96.4	6	5–7	3769	96.0	5	4–7	1353	97.2	5	4–6
Remedial massages	5350	96.1	3	3–4	245	98.8	3	3–4	3785	96.4	3	3–4	1320	94.8	3	3–4
Functional occupational therapies (groups)	5307	95.3	5	4–7	240	96.8	5	4–7	3760	95.7	6	4–7	1307	93.9	5	3–7
Educational presentations—motivation and lifestyle modification	5229	93.9	2	1–3	230	92.7	2	1–2	3692	94.0	2	1–3	1307	93.9	2	1–3
Medical training therapy—resistance training	5168	92.8	5	4–7	244	98.4	6	4–7	3746	95.4	5	4–7	1178	84.6	5	4–6
Social counseling	4870	87.5	2	1–2	233	94.0	2	2–3	3559	90.6	2	1–3	1078	77.4	2	1–2
Educational lectures	4674	84.0	1	1–2	206	83.1	1	1–1	3269	83.2	1	1–1	1199	86.1	1	1–2
Occupational therapy (individual treatment)	4666	83.8	2	2–3	201	81.0	2	2–3	3292	83.8	2	2–3	1173	84.3	3	2–3
Thermotherapy	4248	76.3	4	3–7	180	72.6	4	3–6	3026	77.1	4	3–7	1042	74.9	5	3–7
Electrotherapy	2863	51.4	4	3–6	123	49.6	4	3–6	2067	52.6	4	3–6	673	48.3	5	3–6
Hydrogymnastics	2554	45.9	3	2–4	156	62.9	3	2–4	1905	48.5	3	2–4	493	35.4	3	2–4
Manual lymphatic drainage	1422	25.5	4	3–6	38	15.3	3	2–5	1075	27.4	4	3–6	309	22.2	4	3–6
Cognitive and perception training	1370	24.6	2	2–4	44	17.7	2	1–2	825	21.0	2	1–4	501	36.0	2	2–4
Psychological counseling: sexual therapy	1098	19.7	1	1–4	63	25.4	1	1–3	865	22.0	1	1–4	170	12.2	2	1–4
Inhalation therapies	887	15.9	5	4–7	24	9.7	4	3–6	596	15.2	5	4–7	267	19.2	5	4–7
Creative therapies	597	10.7	2	2–4	33	13.3	4	2–4	439	11.2	2	2–4	125	9.0	2	2–4
Speech therapy	402	7.2	4	2–6	14	5.6	4	1–6	291	7.4	4	2–6	97	7.0	4	2–5
Therapeutic ultrasound	313	5.6	3	2–4	4	1.6	2	2–3	228	5.8	3	2–5	81	5.8	3	2–4

Mdn: median number of treatments per treated patient with interquartile range (IQR). N: number of patients who received specific treatment at least once (percentage of patients of the whole collective). Shown are therapies during the rehabilitative measures of younger patients (18–39 years of age), adult patients of middle age (40–69 years), and elderly patients (70 years and older).

**Table 3 cancers-15-01637-t003:** EORTC QLQ-C30 and HADS mean scores before and after rehabilitation, stratified by age group.

	Age Group	N	T1			T2		Delta	*d*	
			Mean	(SD)	*p* ^a^	Mean	(SD)			*p* ^b^
**Functioning scales (higher scores indicate better functioning)**										
Physical functioning	<40 years	248	77.4	(17.6)	n.s.	84.8	16.1	7.3	0.43	<0.001
	40–69 years	3927	76.0	(19.8)	-	81.8	18.6	5.7	0.30	
	≥70 years	1392	65.4	(22.7)	**	73.9	20.4	8.5	0.40	
Role functioning	<40 years	248	57.1	(29.8)	n.s.	69.8	27.5	12.8	0.45	n.s.
	40–69 years	3927	59.0	(29.8)	-	72.1	26.5	13.1	0.47	
	≥70 years	1392	55.6	(31.1)	n.s.	70.7	27.0	15.1	0.52	
Social functioning	<40 years	248	55.1	(30.4)	n.s.	69.4	29.8	14.2	0.47	0.036
	40–69 years	3927	59.7	(29.5)	-	74.4	26.2	14.8	0.53	
	≥70 years	1392	58.7	(31.3)	n.s.	76.5	26.0	17.8	0.62	
Emotional functioning	<40 years	248	54.9	(25.6)	n.s.	74.7	22.6	19.8	0.82	n.s.
	40–69 years	3927	57.8	(25.0)	-	77.3	22.2	19.4	0.82	
	≥70 years	1392	59.4	(24.7)	n.s.	78.2	21.2	18.9	0.82	
Cognitive functioning	<40 years	248	70.2	(28.7)	n.s.	77.2	23.8	7.1	0.27	n.s.
	40–69 years	3927	73.3	(26.4)	-	77.5	24.1	4.3	0.17	
	≥70 years	1392	74.1	(24.9)	n.s.	78.8	22.6	4.7	0.20	
Global health/QOL	<40 years	248	61.4	(17.0)	n.s.	74.4	16.5	13.0	0.78	n.s.
	40–69 years	3927	59.3	(19.1)	-	74.3	16.9	15.0	0.83	
	≥70 years	1392	54.7	(19.9)	**	70.7	17.7	16.1	0.85	
**Symptom scales (higher scores indicate higher impairment)**										
Fatigue	<40 years	248	54.4	(26.8)	n.s.	38.8	(23.0)	15.5	0.62	n.s.
	40–69 years	3927	50.8	(25.8)	-	35.6	(23.9)	15.1	0.61	
	≥70 years	1392	56.1	(26.5)	**	39.3	(24.1)	16.9	0.67	
Pain	<40 years	248	34.9	(27.7)	n.s.	24.9	(23.2)	10.0	0.39	<0.001
	40–69 years	3927	39.1	(28.6)	-	29.4	(25.9)	9.7	0.35	
	≥70 years	1392	42.7	(30.4)	n.s.	29.7	(28.0)	13.0	0.44	
Nausea/vomiting	<40 years	248	11.8	(20.7)	n.s.	8.1	(19.7)	3.7	0.18	n.s.
	40–69 years	3927	10.9	(19.7)	-	5.4	(14.1)	5.5	0.32	
	≥70 years	1392	12.7	(22.1)	n.s.	7.0	(17.1)	5.7	0.29	
Sleep disturbances	<40 years	248	39.7	(34.7)	n.s.	34.7	(32.8)	5.0	0.15	<0.001
	40–69 years	3927	46.3	(33.4)	-	38.9	(32.6)	7.4	0.22	
	≥70 years	1392	46.7	(34.8)	n.s.	35.8	(32.7)	10.9	0.32	
Dyspnea	<40 years	248	32.3	(32.8)	n.s.	24.6	(25.8)	7.7	0.26	n.s.
	40–69 years	3927	30.7	(30.9)	-	28.7	(28.3)	2.0	0.07	
	≥70 years	1392	36.7	(33.6)	**	33.3	(31.0)	3.4	0.10	
Appetite loss	<40 years	248	21.2	(29.9)	n.s.	12.8	(25.3)	8.5	0.31	<0.001
	40–69 years	3927	19.9	(29.3)	-	11.5	(23.1)	8.4	0.32	
	≥70 years	1392	28.9	(35.2)	**	17.1	(29.2)	11.9	0.37	
Constipation	<40 years	248	15.9	(28.4)	n.s.	7.8	(20.8)	8.1	0.32	n.s.
	40–69 years	3927	17.2	(28.2)	-	10.1	(22.3)	7.2	0.28	
	≥70 years	1392	22.9	(32.0)	**	15.3	(27.6)	7.7	0.26	
Diarrhea	<40 years	248	15.1	(25.0)	n.s.	12.9	(23.7)	2.2	0.09	n.s.
	40–69 years	3927	14.4	(26.1)	-	10.1	(21.6)	4.2	0.18	
	≥70 years	1392	18.6	(29.2)	**	12.8	(24.9)	5.8	0.21	
Financial impact	<40 years	248	37.8	(33.4)	**	28.8	(32.3)	9.0	0.27	n.s.
	40–69 years	3927	29.6	(34.0)	-	21.6	(29.9)	8.0	0.25	
	≥70 years	1392	19.5	(29.0)	**	13.5	(24.5)	6.0	0.23	
**HADS (higher scores indicate higher psychological distress)**										
Anxiety	<40 years	248	6.8	(4.0)	n.s.	5.1	(3.5)	1.6	0.44	<0.001
	40–69 years	3927	6.7	(3.9)	-	4.7	(3.4)	2.0	0.54	
	≥70 years	1392	5.8	(3.6)	**	4.2	(3.2)	1.6	0.47	
Depression	<40 years	248	4.9	(3.4)	n.s.	3.1	(2.9)	1.8	0.56	n.s.
	40–69 years	3927	5.5	(3.9)	-	3.4	(3.3)	2.1	0.59	
	≥70 years	1392	6.0	(3.8)	**	3.9	(3.2)	2.1	0.60	

*p* ^a^: comparison of baseline scores (T1) with age group 40–69 as reference group (indicated by ‘-’) via Tukey HSD test; n.s.: not significant; **: significant at *p* < 0.001 after correction for multiple testing; *d*: Cohen’s *d*. mean differences (delta) on the EORTC QLQ-C30 between T1 and T2 are color coded according to the EORTC QLQ-C30 change scores by Cocks et al. [56] (no color: trivial; yellow: at least a small improvement; green: at least a medium improvement). Mean differences (delta) on the HADS between T1 and T2 are color coded according to the cut-off levels for clinical relevance (i.e., for anxiety: 1.3 points; for depression: 1.4 points) [58]; *p* ^b^: significance test for time × group comparison.

**Table 4 cancers-15-01637-t004:** EORTC QLQ-C30 and HADS mean scores before and after rehabilitation, stratified by Norton scale risk score.

			T1			T2				
	Norton Score	N	Mean	(SD)	*p* ^a^	Mean	(SD)	Delta	*d*	*p* ^b^
**Functioning scales (higher scores indicate better functioning)**										
Physical functioning	Low risk	4140	76.6	18.8	**	82.5	(17.4)	5.9	0.33	<0.001
	High risk	566	53.4	23.1		63.0	(22.8)	9.6	0.42	
Role functioning	Low risk	4140	60.5	29.3	**	73.7	(25.6)	13.2	0.48	n.s.
	High risk	566	44.4	31.6		58.0	(30.1)	13.6	0.44	
Social functioning	Low risk	4140	61.2	29.2	**	75.8	(25.6)	14.5	0.53	<0.001
	High risk	566	48.2	32.0		67.7	(29.3)	19.5	0.64	
Emotional functioning	Low risk	4140	59.2	24.7	**	78.3	(21.4)	19.1	0.83	n.s.
	High risk	566	52.0	25.4		71.9	(24.2)	19.9	0.80	
Cognitive functioning	Low risk	4140	74.3	25.9	**	78.9	(23.2)	4.6	0.19	n.s.
	High risk	566	68.1	27.2		72.3	(26.1)	4.2	0.16	
Global health/QOL	Low risk	4140	60.1	18.7	**	74.8	(16.5)	14.7	0.83	n.s.
	High risk	566	47.7	19.9		64.4	(18.4)	16.8	0.87	
**Symptom scales (higher scores indicate higher impairment)**										
Fatigue	Low risk	4140	50.0	25.7	**	34.8	(23.2)	−15.2	0.62	n.s.
	High risk	566	65.8	24.0		48.9	(24.5)	−16.8	0.69	
Pain	Low risk	4140	37.7	28.2	**	27.7	(25.2)	−10.0	0.37	n.s.
	High risk	566	51.3	31.5		38.1	(30.7)	−13.2	0.42	
Nausea/vomiting	Low risk	4140	10.3	19.2	**	5.3	(14.2)	−5.1	0.30	0.042
	High risk	566	17.8	26.0		10.1	(19.9)	−7.8	0.34	
Sleep disturbances	Low risk	4140	45.0	33.7	**	37.7	(32.7)	−7.3	0.22	n.s.
	High risk	566	51.3	34.7		39.9	(34.0)	−11.3	0.33	
Dyspnea	Low risk	4140	30.2	30.7	**	28.3	(28.0)	−2.0	0.07	0.015
	High risk	566	44.4	35.6		38.7	(33.9)	−5.7	0.16	
Appetite loss	Low risk	4140	20.2	29.7	**	11.8	(23.8)	−8.4	0.31	n.s.
	High risk	566	33.5	35.8		21.5	(31.5)	−11.9	0.35	
Constipation	Low risk	4140	17.0	28.1	**	10.3	(22.6)	−6.8	0.27	n.s.
	High risk	566	27.1	34.1		17.6	(30.0)	−9.5	0.30	
Diarrhea	Low risk	4140	14.5	26.2	**	10.2	(21.7)	−4.3	0.18	n.s.
	High risk	566	21.7	30.3		13.8	(25.7)	−7.9	0.28	
Financial impact	Low risk	4140	27.0	33.0	**	19.6	(28.7)	−7.3	0.24	n.s.
	High risk	566	29.9	33.7		23.3	(32.2)	−6.6	0.20	
**HADS (higher scores indicate higher psychological distress)**										
Anxiety	Low risk	4140	6.4	(3.8)	**	4.5	(3.4)	1.9	0.53	n.s.
	High risk	566	6.6	(3.9)		5.0	(3.6)	1.7	0.44	
Depression	Low risk	4140	5.3	(3.7)	**	3.3	(3.1)	2.1	0.60	n.s.
	High risk	566	7.1	(4.2)		4.9	(3.7)	2.2	0.57	

*p* ^a^: comparison of baseline scores (T1); n.s.: not significant; **: significant at *p* < 0.001 after correction for multiple testing; *d*: Cohen’s *d*. mean differences (delta) between T1 and T2 are color coded according to the EORTC QLQ-C30 change scores by Cocks et al. [56] (no color: trivial; yellow: at least a small improvement; green: at least a medium improvement). Mean differences (delta) on the HADS between T1 and T2 are color coded according to the cut-off levels for clinical relevance (i.e., for anxiety: 1.3 points; for depression: 1.4 points) [58]. *p* ^b^: significance test for time × group comparison.

**Table 5 cancers-15-01637-t005:** EORTC QLQ-C30 and HADS mean scores before and after rehabilitation, stratified by sex.

			T1			T2				
	Sex	N	Mean	SD	*p* ^a^	Mean	SD	Delta	*d*	*p* ^b^
**Functioning scales (higher scores indicate better functioning)**										
Physical functioning	Male	2077	73.4^.^	21.7	n.s.	80.6	(19.8)	7.2	0.35	n.s.
	Female	3490	73.5	20.6		79.6	(19.0)	6.1	0.31	
Role functioning	Male	2077	58.3^.^	31.4	n.s.	71.3	(26.7)	13.0	0.45	n.s.
	Female	3490	57.9	29.5		71.9	(26.7)	13.9	0.50	
Social functioning	Male	2077	57.6	30.6	*	73.3	(26.4)	15.7	0.55	n.s.
	Female	3490	60.2	29.7		75.6	(26.3)	15.4	0.55	
Emotional functioning	Male	2077	61.0	24.5	**	78.3	(21.4)	17.3	0.75	<.001
	Female	3490	56.3	25.1		76.9	(22.2)	20.5	0.87	
Cognitive functioning	Male	2077	76.1	25.1	**	79.6	(23.0)	3.5	0.15	n.s.
	Female	3490	71.7	26.6		76.8	(24.1)	5.1	0.20	
Global health/QOL	Male	2077	56.9	19.9	**	71.5	(17.6)	14.6	0.78	n.s.
	Female	3490	59.1	19.0		74.6	(16.8)	15.5	0.87	
**Symptom scales (higher scores indicate higher impairment)**										
Fatigue	Male	2077	49.7	26.5	**	34.8	(23.7)	−14.9	0.59	n.s.
	Female	3490	53.8	25.8		37.8	(24.0)	−16.0	0.64	
Pain	Male	2077	37.1	29.1	**	26.5	(25.9)	−10.6	0.39	n.s.
	Female	3490	41.4	29.0		30.9	(26.4)	−10.5	0.38	
Nausea/vomiting	Male	2077	10.7	19.9	n.s.	5.8	(14.7)	−4.9	0.28	n.s.
	Female	3490	11.8	20.6		6.0	(15.4)	−5.7	0.31	
Sleep disturbances	Male	2077	40.5	33.8	**	32.2	(31.9)	−8.3	0.25	n.s.
	Female	3490	49.4	33.5		41.3	(32.7)	−8.1	0.24	
Dyspnea	Male	2077	30.7	31.7	*	27.3	(29.0)	−3.4	0.11	n.s.
	Female	3490	33.2	31.8		31.0	(28.9)	−2.1	0.07	
Appetite loss	Male	2077	24.8	32.6	**	14.8	(26.4)	−10.0	0.34	n.s.
	Female	3490	20.7	30.1		11.9	(24.0)	−8.8	0.32	
Constipation	Male	2077	16.9	27.4	**	10.1	(22.1)	−6.8	0.27	n.s.
	Female	3490	19.6	30.3		12.0	(24.7)	−7.6	0.28	
Diarrhea	Male	2077	17.7	27.6	**	12.3	(23.3)	−5.3	0.21	n.s.
	Female	3490	14.1	26.3		10.1	(22.1)	−4.0	0.17	
Financial impact	Male	2077	26.1	32.4	n.s.	19.4	(28.5)	−6.8	0.22	n.s.
	Female	3490	28.3	33.5		20.2	(29.3)	−8.0	0.26	
**HADS (higher scores indicate higher psychological distress)**										
Anxiety	Male	2077	5.9	(3.7)	**	4.3	(3.4)	1.6	0.46	<0.001
	Female	3490	6.8	(3.8)		4.8	(3.4)	2.0	0.56	
Depression	Male	2077	5.7	(3.9)	n.s.	3.7	(3.3)	2.0	0.55	n.s.
	Female	3490	5.5	(3.8)		3.4	(3.2)	2.1	0.61	

*p* ^a^: comparison of baseline scores (T1); n.s.: not significant; *: significant at *p* < 0.05 after correction for multiple testing; **: significant at *p* < 0.001 after correction for multiple testing; *d*: Cohen’s *d*. mean differences (delta) between T1 and T2 are color coded according to the EORTC QLQ-C30 change scores by Cocks et al. [56] (no color: trivial; yellow: at least a small improvement; green: at least a medium improvement. Mean differences (delta) on the HADS between T1 and T2 are color coded according to the cut-off levels for clinical relevance (i.e., for anxiety: 1.3 points; for depression: 1.4 points) [58]. *p* ^b^: significance test for time × group comparison.

## Data Availability

The data of the study are available from the corresponding author upon reasonable request.

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
