# Peer review of "Associations of Age and Sex with the Efficacy of Inpatient Cancer Rehabilitation: Results from a Longitudinal Observational Study Using Electronic Patient-Reported Outcomes"

_cancers, 2023, doi:10.3390/cancers15061637_

Round 1
Reviewer 1 Report
Thanks for the opportunity to review your work entitled “Association of age and sex with the efficacy of inpatient cancer rehabilitation: Results from a longitudinal observational study using electronic patient-reported outcomes.” While I find your paper overall nicely written and easy to follow, I have the following major concerns regarding your paper, which hindered my full support of your paper. I hope the authors find the following comments helpful.
- You did a reasonable job of introducing the importance of studying age and frailty in relation to health status after cancer treatment, two issues remain insufficiently addressed as of now. First, Why age and frailty would impact the treatment effect/benefit of rehabilitation services and, second, if age and frailty do impact the treatment effect, so what? Age (and arguably frailty) is a difficult-to-modify factor, if at all malleable.
- While I agree with the authors on the unique psychosocial challenges among AYAs with cancer, but the thinking behind it is all the unique developmentally specific challenges associated, but not necessarily age itself, no? This casts doubt for me regarding your selection of age as a “moderating” variable.
- In the paragraph where you talked about sex differences being reported with respect to unmet needs, and you primarily cited empirical studies, what are some of the theoretical rationales behind them? Just because empirical studies reported sex differences is not sufficient ground for investigating the role of sex differences.
- [possibly repeating comment] “In the present investigation, we wished to analyze whether differences of age or sex are associated with different outcomes of oncological rehabilitation…” Again, now to disagree with the authors on the important role of age and sex, but I consider your literature review section falls short in justifying the contribution of your current study, i.e., the so what question. I am not looking for a couple of sentences but a reconceptualization of your literature review section to make a stronger argument.
- Also, what about frailty, which was mentioned in your literature review section, but I don’t see it being included in your study purpose in your literature section?
- Psychometric properties, i.e., Cronbach’s alpha were not reported for the measures used in your study, at least, I don’t see them in your measurement section.
- In your method section, the authors talked about “clinically meaningful change.” I assumed that the authors were referring to the concept of Clinically Meaningful Important Difference (i.e., CMID). However, studies cited or referenced in your measurements do not reflect that but something else. For EORTC QLQ-C30, for example, the authors may consider the study: “Minimal clinically meaningful differences for the EORTC QLQ-C30 and EORTC QLQ-BN20 scales in brain cancer patients” by J. Maringwa and colleagues published in 2011. And for HADS, the authors may consider the study: “The minimal important difference of the hospital anxiety and depression scale in patients with chronic obstructive pulmonary disease” by Puhan and colleagues in 2008, OR, the study: “Establishing the Minimal Clinically Important Difference for the Hospital Anxiety and Depression Scale in Patients With Cardiovascular Disease” by Lemay and colleagues in 2019.
- Rationales regarding why the authors used ANOVA versus ANCOVA (Analysis of Covariance) should be briefly discussed and justified. For your ANOVA analysis, any post-hoc analyses or corrections for p-values were considered? If not applicable, you want to briefly discuss why the consideration is not applicable.
- The authors are also encouraged to consider the baseline score as a control variable in ANCOVA, even if your dependent variable is the change score. One may reasonably argue that the baseline score can potentially have an impact on a participant’s change, which should be considered for, no?
- Given the major concerns of your method/analysis, at least how it is currently reported, I did not go through your results as I hold concerns about findings from your current analyses.
Author Response
Thank you for your helpful comments. For a detailed point-to-point reply, please see the attached file

Reviewer 2 Report
Cancer rehabilitation continues to receive insufficient attention compared to diagnosis and early treatment. This then is a strength of the manuscript. It benefited from a large data set with consistency in payment (i.e., access to services) by virtue of the sample’s origins. The authors have published on the topic of cancer rehabilitation previously. In this new manuscript, they consider potential differences by virtue of sex/gender, age, and what the authors refer to as frailty.
The study’s stated aim (which is more clearly articulated on page 16 at the end of the manuscript than at its beginning) is to identify differences in HRQOL and functional health by age and sex and to determine how rehabilitation might restore impairments in certain subgroups of patients (i.e., older and younger subgroups and men and women). My concern is with the potential of the authors’ findings to inform and improve treatment services. The authors tell us that one would expect the need for, and benefit of, cancer rehabilitation to be greater among older patients. This is well-known. Thus, finding that older adults with cancer have greater needs does not push the envelope on our knowledge base. It is not new. Also, the choice to set the upper age limit for AYA to 39 years somewhat confuses interpretation of findings. This is because patients between the ages 20 and 39 years are more likely to be working and to have families of their own. Those in the workforce likely would experience greater stressors to find or maintain employment. Examining patients under 20 years of age might have revealed more in terms of symptom change.
No differences were found by gender. Here I am concerned with age being influenced by cancer type. The authors acknowledge that breast cancer patients are more likely to make up the group of younger women in their study. Impairments and life challenges for these women, who in the UK have a five-year survival rate of 86.3% (compared to 7.3% for cancer of the pancreas and 13% for cancer of the liver), arguably would be different from those of women or men with other cancer types or from older women with breast cancer. In other words, considering all cancers as a group and focusing solely on age and sex or gender might miss the unique challenges of specific cancers and their distinct treatment modalities. Treatment modalities differ by type. Lymphedema treatment among breast cancer patients is but one example. Anxiety and depression are influenced by type and extent of impairment. One size does not fit all.
The most salient finding is somewhat embedded on page 18 (lines 567-569). Here we are told that based on their findings, while all patients benefit from cancer rehabilitation, older adults might profit more from a stronger focus on physiotherapy while younger patients might benefit more from a social worker to help them negotiate financial difficulties/challenges. More discussion of how this might shape clinic planning and functioning would strengthen the manuscript. Are social workers members of all treatment teams, for example, or must they be called in when need is determined? Making them routine members of a rehabilitation team might, for example, be a simple fix.
Author Response

(The authors gave the same response as above.)

Round 2
Reviewer 2 Report
The authors have address most of my concerns. I still worry about the utility for informing care, but it is better. There are some wording issues that can be addressed fairly easily (e.g., line 179 through 181).
Author Response
We wish to thank you for your thorough review of our manuscript in the first round of revisions. We have asked a native English speaker for editing of the paper. With his support, we have corrected several wording issues including the one noted by you.